# Evolutionary rescue by compensatory mutations is constrained by genomic and environmental backgrounds

Marie Filteau, Véronique Hamel, Marie-Christine Pouliot, Isabelle Gagnon-Arsenault, Alexandre K Dubé & Christian R Landry[*]

## Abstract

Since deleterious mutations may be rescued by secondary mutations during evolution, compensatory evolution could identify genetic solutions leading to therapeutic targets. Here, we tested this hypothesis and examined whether these solutions would be universal or would need to be adapted to one's genetic and environmental makeups. We performed experimental evolutionary rescue in a yeast disease model for the Wiskott–Aldrich syndrome in two genetic backgrounds and carbon sources. We found that multiple aspects of the evolutionary rescue outcome depend on the genotype, the environment, or a combination thereof. Specifically, the compensatory mutation rate and type, the molecular rescue mechanism, the genetic target, and the associated fitness cost varied across contexts. The course of compensatory evolution is therefore highly contingent on the initial conditions in which the deleterious mutation occurs. In addition, these results reveal biologically favored therapeutic targets for the Wiskott–Aldrich syndrome, including the target of an unrelated clinically approved drug. Our results experimentally illustrate the importance of epistasis and environmental evolutionary constraints that shape the adaptive landscape and evolutionary rate of molecular networks.

**Keywords** aneuploidy; epistasis; experimental evolution; genotype-by-environment interaction; Wiskott–Aldrich syndrome

**Subject Categories** Chromatin, Epigenetics, Genomics & Functional Genomics

**Mol Syst Biol.** (2015) 11: 832

## Introduction

Genetic and environmental interactions constrain the course of adaptive evolution by limiting the number of available genetic paths to increased fitness (Weinreich *et al*, 2006; Kvitek & Sherlock, 2011; de Vos *et al*, 2013; Chiotti *et al*, 2014; Hartl, 2014). A given adaptive mutation may be advantageous in only a limited set of conditions, which makes its contribution to adaptation dependent on the environment and genotypes of the individuals in which the mutation travels on its way to fixation in the population. The effects of deleterious mutations are also variable across environments and genotypes, as illustrated by the variability of gene essentiality across conditions and genetic backgrounds (Dowell *et al*, 2010). In addition, these genetic (GxG) and genotype-by-environment (GxE) interactions can themselves be dependent upon other factors, revealing higher-order interactions (GxGxE, GxGxG and even, GxGxGxE). For instance, studies using reverse genetics showed that genetic interactions, primarily aggravating interactions, can be sensitive to changes in environmental conditions (GxGxE) (Harrison *et al*, 2007; St Onge *et al*, 2007; Bandyopadhyay *et al*, 2010; Guenole *et al*, 2013; Zhu *et al*, 2014) and also diverge between related and distant species (GxGxSp) (Dixon *et al*, 2008; Roguev *et al*, 2008; Tischler *et al*, 2008). These complex interactions suggest that overcoming the effect of a deleterious mutation by compensatory evolution is most likely subjected to the same constraints as adaptive evolution, as highlighted by a recent report on plastic compensatory mutation effects across environments (GxGxE) (Szamecz *et al*, 2014). Understanding these complex interplays of genetic and environmental effects is one of the great challenges in systems biology (Fischbach & Krogan, 2010; Weinreich *et al*, 2013) because it requires a mechanistic understanding of genotype–phenotype maps (Landry & Rifkin, 2012).

Higher-order interactions are also very important in medical genetics as they make therapies (genetic or pharmaceutical) for human deleterious mutations potentially highly dependent on complex genotype and environmental dependencies (Eichler *et al*, 2010). These interactions might limit the development of treatments for the large number of mutations responsible for genetic diseases being discovered (Koboldt *et al*, 2013), the vast majority of which remain untreatable (Sakharkar *et al*, 2007). In particular, the identification of pharmaceutically relevant targets for loss-of-function genetic diseases that are caused by the impairment of one protein, often with pleiotropic consequences, is challenging (Segalat, 2007). The challenge comes partly from the fact that the remedies must

Département de Biologie, PROTEO and Institut de Biologie Intégrative et des Systèmes (IBIS), Université Laval, Québec, Qc, Canada
*Corresponding author. Tel: +1 418 656 3954; Fax: +1 418 656 7176; E-mail: christian.landry@bio.ulaval.ca

involve the network of genes and proteins associated with the non-functional gene rather than from the gene itself. Compensatory mutations for these deleterious alleles may inform us on how this network may need to change to return to a functional state.

Here, we use an experimental evolution approach to identify a target-oriented network of genetic remedies to a deleterious mutation in a combination of genetic and environmental conditions. We used the budding yeast as a model for the study of the Wiskott–Aldrich syndrome (WAS). WAS is a rare X-linked primary immunodeficiency and blood platelet disorder classically characterized by the triad of recurrent infections, abnormal bleeding caused by a reduced number of platelets, and skin eczema (Albert *et al*, 2011).

The syndrome is caused by mutations in the gene encoding the scaffolding protein WAS (WASP) involved in actin assembly (Thrasher & Burns, 2010). WASP is a functional homolog of *Saccharomyces cerevisiae* Las17 and it can complement the growth defect of the *las17Δ* strain when co-expressed with the WAS-interacting protein (WIP), the yeast Vrp1 homolog (Rajmohan *et al*, 2009). The yeast *las17-41* allele encodes a W41R mutation that is homologous to the W64R mutation in the WH1 domain of WASP (Fig EV1). This mutation causes classic WAS symptoms (Fillat *et al*, 2000; Jin *et al*, 2004). The growth of a yeast strain carrying this allele is completely impaired at 37°C and above, while it is normal at 22°C (Fig EV2). The *las17-41* allele therefore provides a tractable system to study

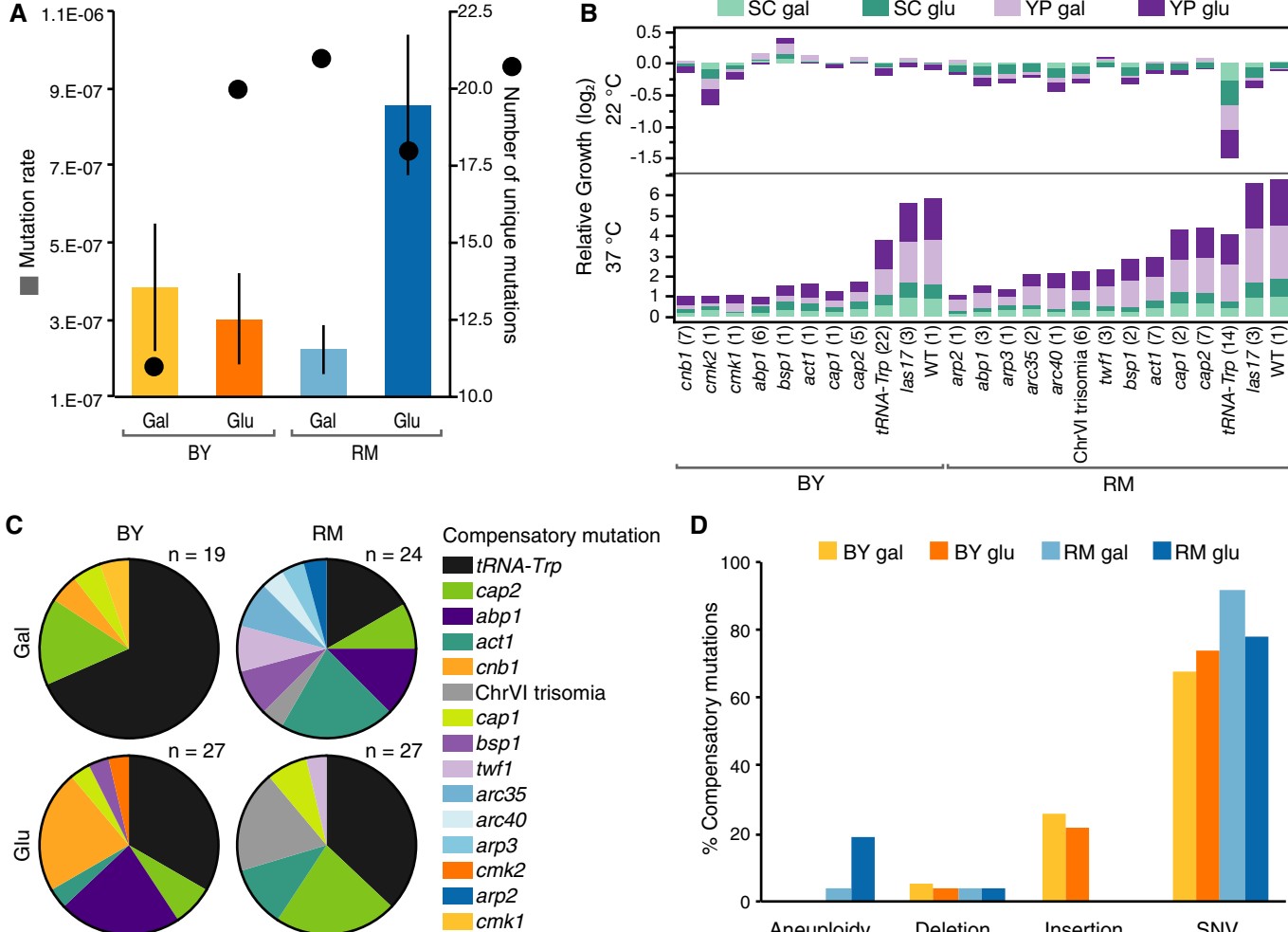

**Figure 1. Context-dependent *las17-41* compensatory evolution.**

A   Compensatory mutation rate of the thermosensitive *las17-41* phenotype in four contexts: RM and BY genetic backgrounds and two carbon sources, glucose (Glu) and galactose (Gal) (left axis, *n* = 3, 8, 4, and 4 for BY gal, BY glu, RM gal, and RM glu, respectively). Bars = average, error bars = SD. Overlaid dots show the number of unique rescue mutation (right axis) found in each context. Mutations at a shared locus and nucleotide position were considered equivalent.

B   The average growth of evolved strains sharing compensatory mutation types at 22°C (top panel) and 37°C (bottom panel) in four culture media is shown relative to their respective progenitor. The *las17* strains carry R41W, R41L, or R41Q functional reversions. WT = wild-type strain. Growth phenotypes were measured on synthetic complete (SC) and rich media (YP), *n* = number of strains next to labels, each inferred from eight replicates. One *cap1* mutant was excluded because of mitochondrial loss and one mutant because it contained two rescue mutations (*act1* and *cap2*).

C   Proportion of compensatory mutations identified in the four experimental contexts. Only two genes were identified in all four contexts, indicating that at the target level, up to 80% of the mutations could be context dependent.

D   Proportion of each rescue mutation type encountered in each experiment, *n* = same as in C, SNV = single nucleotide variant.

how this mutation could be corrected genetically or pharmacologically. We experimentally explored the genetic rescue by compensatory mutation to WAS in four different contexts, using two *las17-41* strains with different genetic backgrounds, referred to as BY and RM, each on two carbon sources, glucose and galactose, which induce distinct metabolic states in yeast (Fendt & Sauer, 2010). We find that the mutation rate and up to 80% of rescue mutation targets could be constrained by the genetic and/or environmental context. We experimentally exemplify that interactions between these factors, that is, higher-order effects, can have critical consequences on treatment outcome.

## Results and Discussion

To reveal potential genotype and environmental effects on the possible compensatory trajectories to the *las17-41* allele, we used fluctuation assays to measure the rate at which the first compensatory mutations occur in these contexts (Fig 1A). The fluctuation assay is a short-term evolution experiment in which small parallel populations of the thermosensitive strains are allowed to accumulate conditionally neutral mutations in synthetic media under permissive conditions (22°C) for around 20 generations. The genetically

variable populations are then placed under selective conditions (37°C). Only the rare cells that pre-acquired a rescue mutation are able to form a colony, and the frequency of these events is recorded at the population level to calculate the compensatory mutation rate. We found a significant strain-by-carbon source effect (linear model, df = 18, whole model *P*-value = 1.9e-5, interaction *P*-value = 5.8e-5), meaning that the combination of genetic and environmental factors influences the compensatory mutation rate. Indeed, there is a higher mutation rate for the ability to rescue growth at 37°C in the RM genetic background on glucose (Tukey HSD, *P*-value ≤ 0.05). We then studied independently evolved mutants isolated from these assays, which are expected to typically contain only one compensatory rescue mutation each.

First, we looked for intramolecular compensatory mutations by sequencing the *LAS17* locus in 330 isolated compensatory mutants. Only 5% of compensatory mutations occurred within *las17-41* itself. Therefore, the reversion rates can not explain the context-dependent compensatory mutation rate. All intramolecular mutations affected the originally mutated residue and we encountered both true reversions (R41W, 11%) and pseudo-reversions (R41L, 44%, R41Q, 44%). Thus, residue 41 is critical for Las17 function and *las17-41* may offer very little in terms of how this protein could itself be genetically or pharmacologically modified to recover its functionality.

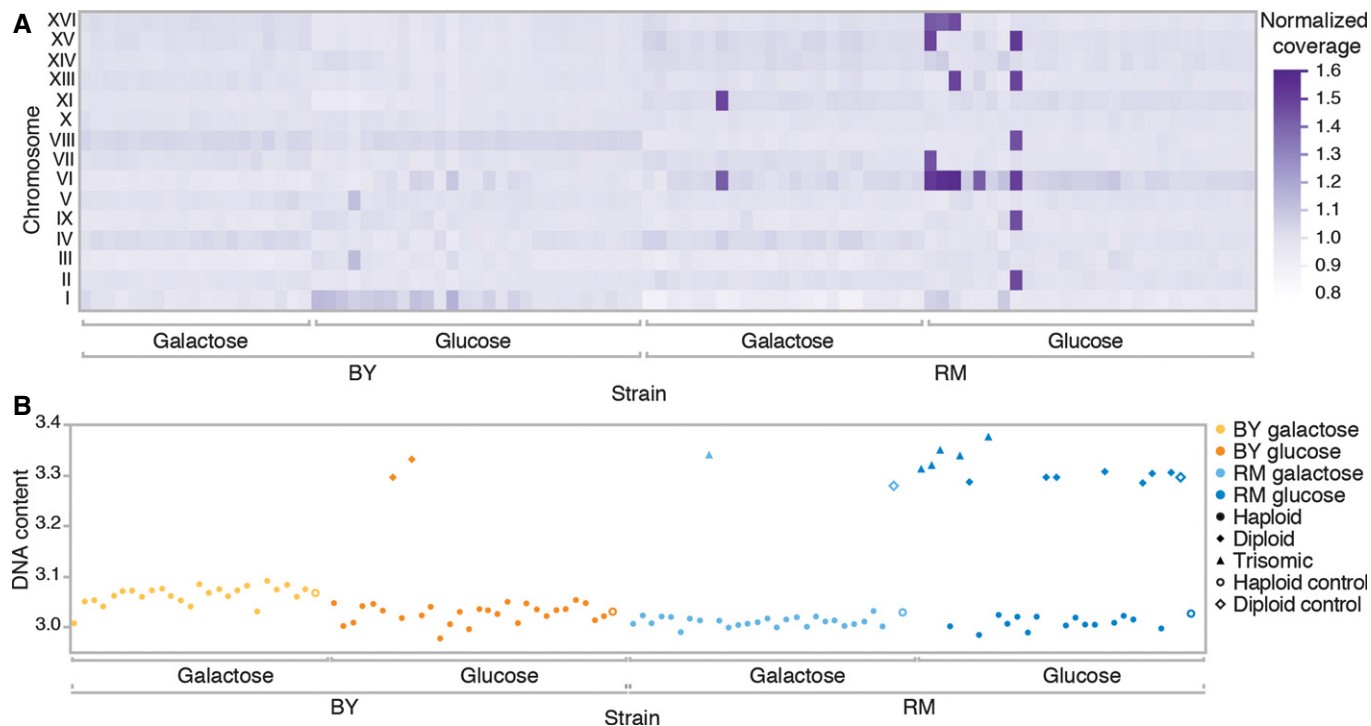

**Figure 2.   Ploidy variation among strains.**

A   Six strains had unequal read coverage across chromosomes, with a coverage of approximately 1.5× along up to six entire chromosomes. This factor is consistent with trisomia in diploid strains. The heat map shows the normalized average coverage across chromosomes for each sequenced mutant strains. ChXII is not shown because of a constant bias in its coverage estimate caused by the multiple copies of rRNA genes it encodes.

B   Ploidy measurements of all the sequenced strains using DNA content measurement by flow cytometry confirmed trisomia. Haploid and diploid assignments were determined by *k*-means clustering on test and control strains. Trisomic assignment is based on the unequal coverage values in (A). A diploid control strain of the RM genetic background is also included. Some strains for which a coding mutation was identified were also diploid. Because the frequency of the variant was in all cases near 100% (Table EV1), meaning that the diploid mutants are homozygous for the rescue mutation, it can be assumed that diploidization occurred after the mutation.

The mutations were considered functional reversions because the fitness of those mutants was comparable to the wild type (Fig 1B). We found that context has no consequences on functional reversion rates (nominal logistic model, df = 3, *P*-value = 0.33).

Second, we identified intermolecular compensatory mutations in a subset of 96 strains using whole genome sequencing. We identified 97 compensatory mutations in 96 strains (Table EV1) enriched for predicted functional effects based on bioinformatics analyses (right-sided *P*-value = 3e-4) (Fig EV3). The number of locus-unique mutations did not correlate with the compensatory mutation rate but differed substantially among contexts, with the largest difference observed between genetic backgrounds in galactose (Fig 1A). Thus, the elevated mutation rate in the RM-glucose condition is apparently not owed to a broader mutational target size of rescue mutations.

The most frequent mutation (36 total occurrences) was a suppressor mutation in the tRNA-Trp anti-codon that restores the Las17 protein sequence by translating the arginine codon CGG as a tryptophan (Table EV1). An increased number of targets can explain the high frequency of this mutation, as there are six nuclear copies of the tRNA-Trp gene in the yeast genome, in each of which we observed individual occurrences of the same suppressor mutation. Interestingly, the frequency of this mutation varied between contexts (nominal logistic model, df = 3, whole model *P*-value = 0.005, interaction *P*-value = 0.005) (Fig 1C), again indicating complex interactions in the favored evolutionary paths.

At the phenotypic level, evolved strains also showed different growth profiles according to their mutation, reflecting that they are not all equally compensatory, and that their effects vary across growth conditions (Fig 1B). Some compensatory mutations show genetic background-specific fitness costs at permissive temperature. For instance, the tRNA-Trp fitness cost is higher in RM than in BY. This could be explained by the fact that this alteration would be more pleiotropic in one background than the other, with about a hundred genes in RM that would not be affected in BY, including 11 essential genes, because the CGG codon occurs more often in RM than in BY (Table EV2).

The type of genetic changes by which compensation occurs is also context dependent. On the one hand, we identified insertions of transposable elements (TE) belonging to the *Ty1* retrotransposon family in the BY genetic background only (Fig 1D). This result can be explained by the lack of active *Ty1* coding element in the RM ancestor as opposed to the BY ancestor that contains 32 full-length copies of *Ty1* coding element (Bleykasten-Grosshans *et al*, 2013). On the other hand, six RM mutants showed various degrees of aneuploidy (Fig 2A) and several mutants became diploid over the course of the experiment (Fig 2B), a phenomenon often observed in experimental evolution (Gerstein *et al*, 2006). However, the aneuploid strains all shared a ChrVI trisomia in a diploid background (Fig 2A and B), an unlikely bias by chance alone (likelihood ratio, *P*-value = 3e-6), indicating that this particular aneuploidy is a compensatory rescue mechanism, or an elevated rate of trisomy specific for this chromosome in this genetic background. Our results are likely owed to the former, given the normally low tolerance of ChrVI copy number variation, most probably due to the many cytoskeletal proteins it encodes such as actin and tubulin (Anders *et al*, 2009; Zhu *et al*, 2012).

Aside from suppressor mutations and reversions, we identified compensatory mutations in 13 protein-coding genes that belong to

major functional modules: the actin filament and actin cortical patch, the ARP2/3 complex, and the Ca²⁺/calmodulin signaling pathway, the latter two being found in specific contexts (Fig 3). We find that the context has a significant effect on the functional targets recovered (nominal logistic model, df = 9, *P*-value = 7e-6). To investigate the specificity of the identified compensatory mutations, selected alleles were cloned on a low copy number plasmid and transformed into the thermosensitive strains. Growth of transformed colonies at 37°C would confirm that the compensatory mechanism consists in a dominant gain of function. This was the case for most alleles tested, but the effect of each individual mutation was variable among contexts, sometimes even within the same gene (Fig 4A). For example, on glucose, a clear gain of function could be observed for only one *cap1* allele in RM, where it was recovered, but no clear gain of function was observed in BY. In line with their context-specific recovery, alleles of genes in the ARP2/3 complex were consistently compensatory only in the RM background on galactose. This again supports that compensatory rescue depends on higher-order interactions and also suggests that mutations involving the same gene may not compensate via the same mechanism, that is, gain of function at the protein sequence level or gene

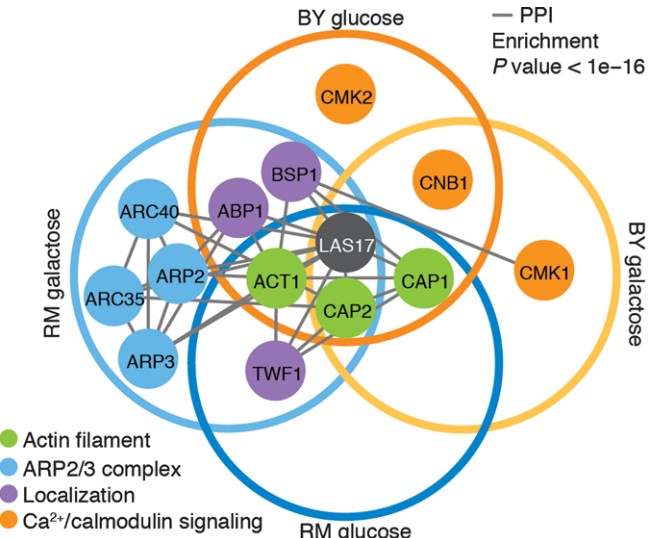

**Figure 3. Network of protein-coding rescue genes.**
Empty circles regroup targets that were found in a particular context. Nodes are colored by their function and edges represent protein–protein interactions reported in Biogrid 3.3.122 (Chatr-Aryamontri *et al*, 2015). Protein-coding compensatory genes are strongly associated with Las17 at the network level as they show 15-fold enrichment in protein–protein interactions among themselves (Fisher's exact test, *P*-value < 1e-16). Ten of these proteins physically interact with Las17, a significant enrichment (Fisher's exact test, *P*-value = 3e-14), suggesting that interacting proteins are prime candidates for rescue mutations. Interestingly, in case where an interaction interface with Las17 has been identified, the mutations found do not coincide with the interacting residues (Chereau *et al*, 2005; Ti *et al*, 2011), suggesting more complex compensatory mechanisms than direct physical interactions with the altered interface of Las17. These interactors also have a smaller shortest path to one another than expected by randomly sampling the Las17 interactome (left-sided *P*-value = 0.03), showing that this subset is intimately connected to the essential function of *LAS17*. In agreement with this observation, these 13 genes are enriched in GO terms related to Las17 functions, for example, actin filament polymerization (GO:0030041, Holm–Bonferroni corrected *P*-value = 3e-14).

dosage. The actin gene is a prime example supporting this conclusion, as increasing gene dosage of actin with a wild-type allele was able to compensate *las17-41* thermosensitivity specifically in the RM background on glucose, whereas some point mutations in actin were compensatory in all contexts (Fig 4A and B). Thus, the increased gene dosage of actin may be the compensatory mechanism explaining the high frequency of ChrVI trisomia. This result parallels the report that the loss of function of the WIP homolog Vrp1 can be suppressed by increased amounts of monomeric actin, but not filamentous actin (Haarer *et al*, 2013). In humans, the WH1 domain where the W41R mutation lies interacts with WIP, and loss of WASP–WIP complex activity has been hypothesized to be causal to the disease (Rajmohan *et al*, 2009). However, we did not encounter compensatory mutations in *VRP1*, suggesting that the interaction could either not be restored in a single step or may not contribute to Las17 essential function at restrictive temperature.

Apart from *BSP1*, all yeast compensatory genes have at least one human homolog (RIDDLE, FDR = 0.007), expanding the associated WAS disease pathway and illustrating the power of such experiments in identifying functionally related proteins (Fig 5). Eight of the genes are likely affected by a partial or complete loss of gene function (deletion, TE insertion, frame shift, or stop codon). We investigated complete loss-of-function specificity by gene deletion in the progenitor strain and confirmed that three deletions (*cnb1Δ*, *bsp1Δ*, and *twf1Δ*) rescued growth of the *las17-41* strain at 37°C, each in a context-specific manner (Fig EV4). Genes whose absence confers some benefit could be attractive drug targets (Kaiser, 2014). For example, the mutations affecting the calcineurin regulatory subunit *CNB1* suggest that calcineurin inhibition compensates

the *las17-41* phenotype in the BY background. WAS and calcineurin are both involved in controlling TCR-mediated T-cell function, but since they are depicted as acting in parallel paths (Baniyash, 2004), their interconnection remains to be explored. Calcineurin function can be inhibited using cyclosporin A, an immunosuppressive drug (Singh-Babak *et al*, 2012). We found that cyclosporin A allows growth of BY *las17-41* at 37°C on glucose (Figs 6 and EV5) which mirrors the *las17-41 cnb1Δ* phenotypes (Fig EV4) and the quantitative results obtained for *cnb1* rescue mutants (Fig 1C). Thus, the genetic background and metabolic state may influence the effect of a potential drug. Therefore, taking these factors into account when conducting screens to identify therapeutic targets may increase their success rate.

Substantial higher-order epistasis is common in nature and has been shown to constrain adaptive evolution (Kvitek & Sherlock, 2011; de Vos *et al*, 2013; Weinreich *et al*, 2013; Hartl, 2014). Here, we show how experimental evolution in a model organism can be used to generate target-oriented networks of genetic remedies for disease mutations and that these networks depend on the genetic background and the environment. When applied to *LAS17*, this approach brings further insights into the essential role of Las17/WAS WH1 domain for which little is known beyond its binding affinity to Vrp1/WIP (Zettl & Way, 2002; Cotta-de-Almeida *et al*, 2015) and highlights connections to other immune-related homologs. The results reveal potential therapeutic targets for pharmaceutical modulation of WASP-mediated signaling, which could evolve as a rational strategy. However, although the BY and RM *las17-41* strains had a similar phenotype, their respective genetic network underpinning the actin cytoskeleton regulation appears to

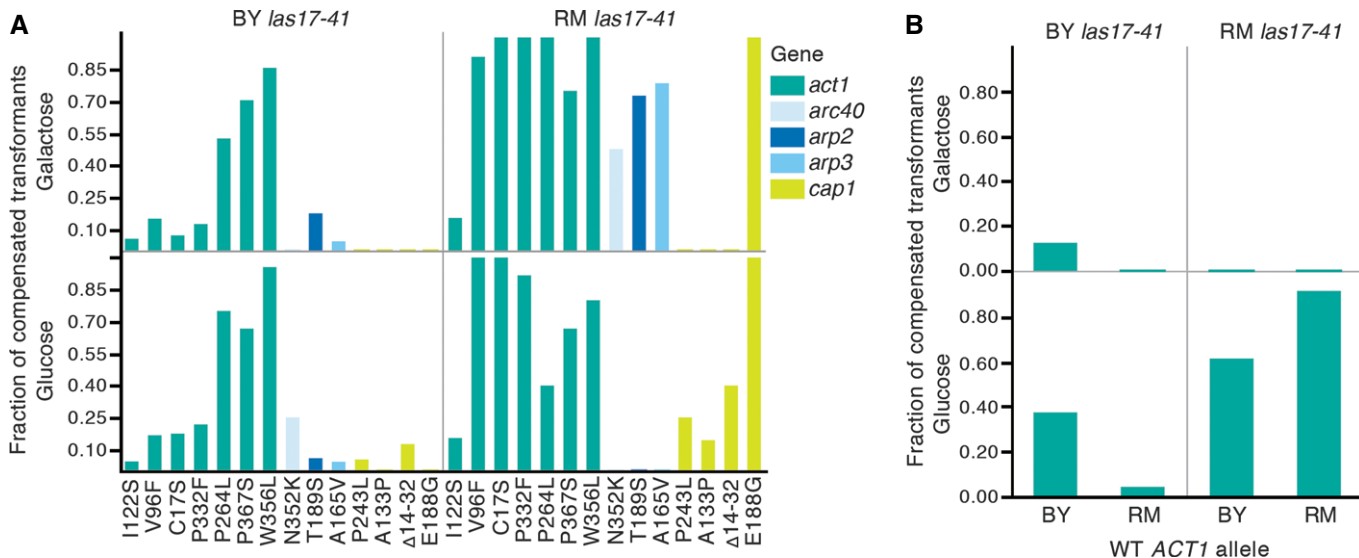

**Figure 4.** **Variation in intra- and intergenic compensatory gain of function across genotypes and environmental conditions.**

A   Bars show the fraction of colonies transformed with a mutant allele showing more growth at 37°C than 95% of the control, that is, the originating wild-type allele in BY and RM *las17-41* thermosensitive strains on glucose and galactose. The *x*-axis indicates the amino acid changes for each allele. A low fraction of compensated transformants could reflect that the addition of the mutant allele increases the compensatory mutation rate only or that it has a partially recessive effect. Because the wild-type copy is also present in the transformed haploid strains, a recessive gain of function would not be revealed by this experiment.

B   Increasing the gene dosage of actin by adding a wild-type actin allele from BY or RM (*x*-axis) can compensate *las17-41* thermosensitivity in the RM background on glucose. Bars show the fraction of colonies that grow more than 95% of control colonies at 37°C. Control colonies are BY and RM *las17-41* transformed with an empty plasmid (pRS316), which reflects the background growth owed to compensatory mutants arising during the experiment.

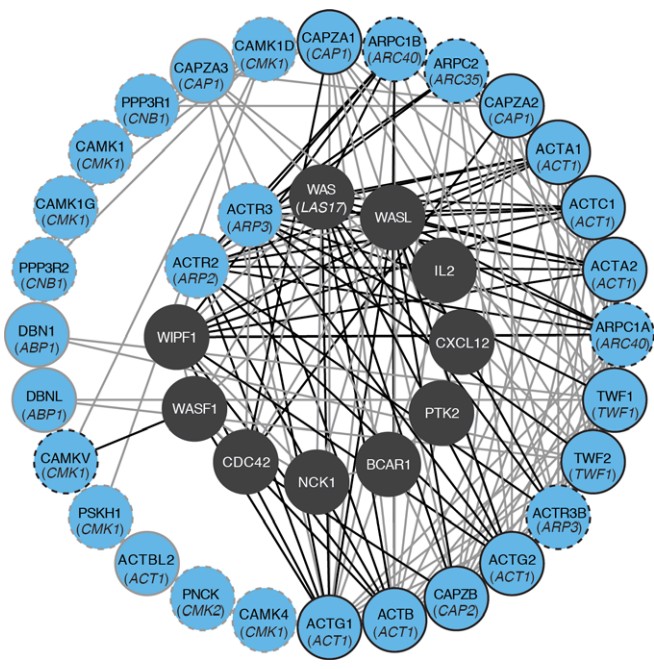

**Figure 5.  A target-oriented network of genetic remedies to WAS disease.**
The inner circle depicts the WAS disease-associated gene network known in human. The blue nodes expanding this network represent human orthologs of *las17-41* compensatory rescue genes (corresponding yeast genes are in parenthesis). HumanNet-based connections between WAS disease genes and compensatory orthologs are highlighted in black. Black outlined nodes are new candidates for human Wiskott–Aldrich syndrome pathway genes. Dashed outlines indicate the orthologous ARP2/3 complex and Ca$^{2+}$/calcineurin pathway, which appear as context-dependent functional target in yeast. The network was generated by MORPHIN (Hwang *et al*, 2014) and visualized with Cytoscape (Su *et al*, 2014).

Source data are available online for this figure.

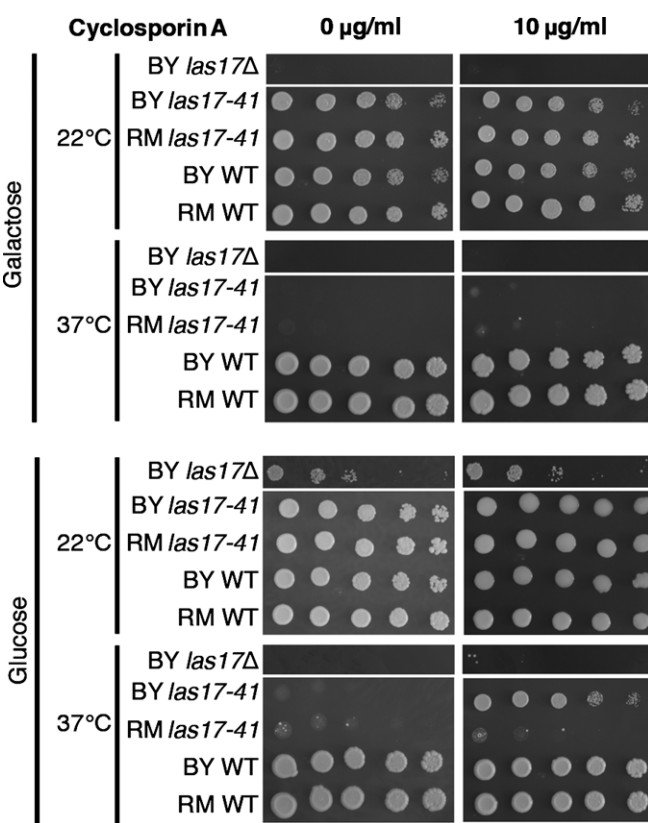

**Figure 6.  Pharmacological inhibition of calcineurin by cyclosporin A compensates the *las17-41* thermosensitivity at 37°C in a genetic background and carbon-source specific manner.**
Optimal growth is observed with 10 µg/ml of cyclosporin A (Fig EV5) indicating that a calcineurin-independent cytotoxic effect of cyclosporin A is likely at play at higher concentrations (Singh-Babak *et al*, 2012), although this would be *las17-41*-specific since the wild-type strain exhibits normal growth. Dissection of heterozygous *LAS17* deletion strains revealed that in glucose, *LAS17* is essential in RM, but thermosensitive in BY as shown by the spot assays. In galactose, *LAS17* is also essential in the BY background. Cyclosporin A does not compensate thermosensitivity of BY *las17Δ* at 37°C, suggesting that its effect is specific to this disease mimicking mutation in *LAS17*. Spot assays shown after 3 days of growth.

have diverged substantially since they last shared a common ancestor, providing a prime example of developmental system drift (True & Haag, 2001; Verster *et al*, 2014). This genetic network diversity could also exist among human populations and the observed context-dependent effects could translate into different efficiencies of therapeutic treatments and different severity in terms of side effects. Moreover, we find that epistatic and environmental factors affect compensatory mutation rate, types, and mechanisms. In natural populations that suffer from limited effective population sizes, compensatory evolution in response to the fixation of a given deleterious mutation may therefore be highly contingent on the initial population genetic composition. This phenomenon would carry important consequences for our understanding of resistance to antibiotic and cancer evolution in which deleterious mutations accumulate and are compensated by further mutations (Bjorkman *et al*, 2000; Ashworth *et al*, 2011). Consequently, the successions of mutations may not only be canalized toward specific paths in individuals of different genetic background, but may also occur at different rates, influencing cancer development speed. Clearly, a better understanding of genetic and environmental interactions may help improve our understanding of evolution as well as complex diseases and their treatments (Ashworth *et al*, 2011).

## Materials and Methods

### Strains

Strains originating from two well-described genetic backgrounds (~0.5% divergence) were used: One, designated as "BY," is a haploid laboratory strain, formally named BY4741, and the other, designated as "RM," is known as RM11-1a. Table EV3 describes the strains used in this study. We sequenced the *las17-14* allele from the thermosensitive allele collection (in BY background—plate 3, position D2) (Li *et al*, 2011) and found that the sequence did not match the previously reported mutations (W41E, L133S) (Barker *et al*, 2007). Thus, we named this allele *las17-41* (W41R, L133S, V362A). We used this BY *las17-41* strain to amplify the las17-41-KanMX cassette and transformed it into AKD0057 to generate the RM *las17-41* strain. The allele sequence was confirmed by Sanger sequencing.

**Culture media**

Complex media (YP) used consisted of 1% yeast extract, 2% tryptone, and 2% glucose or galactose, while complete synthetic media (SC) was made with 0.175% yeast nitrogen base, 2% glucose or galactose, 0.135% complete amino acid dropout or 0.172% –URA amino acid dropout, and 0.1% MSG. A total of 2% agar was added to obtain solid medium. When needed, antibiotics were added at the following concentration: 200 mg/ml G418 (Bioshop) and 100 mg/ml clonNAT (Werner BioAgents).

**Fluctuation assays**

The fluctuation assay was adapted from Lang and Murray (2008). The assays were repeated with eight different clones of each strain. For each experiment, a thermosensitive colony was diluted in 40 ml of SC medium with 2% glucose or galactose. The number of cell per μl was estimated three times by flow cytometry with a guava easy-Cyte 8HT cytometer (Millipore) and 30 μl aliquots containing around 5,000 cells each were placed in 46 wells of a 96-well PCR plate, avoiding corner wells. After 3 days of incubation at room temperature (~20 generations), the whole content of 40 wells was manually spotted on an OmniTray and placed at 37°C for 7 days. Three pairs of wells were then pooled to estimate cell concentration by flow cytometry. The spots with no visible colonies after 7 days were reported as zero-class events, and the following formula was used to calculate the compensatory mutation rate for assays where the fraction of zero-class events was between 10 and 80%:

$$\mu = -\ln(P0)/N$$

where P0 is the probability that a mutation does not occur in the entire culture (zero-class events divided by the total number of cultures tested). $N$ is the number of cell divisions that have occurred, that is, the total number of cells plated minus the total number of cells in the initial inoculum and μ is the mutation rate, also called the mutations of interest per cell division.

One colony was isolated per spot to generate a compensatory mutant collection.

**Reversion rates**

The *las17-41* partial sequences of 330 compensatory mutants were obtained by capillary sequencing (IBIS sequencing platform) to identify possible reversion events.

**Whole genome sequencing**

We performed whole genome sequencing of a subset of 102 compensatory mutant strains and the progenitor strains. For each strain, DNA of 1 ml of an overnight culture grown at 22°C in the carbon source from which they were isolated was extracted (DNeasy blood and tissue kit, Qiagen). Growth at 37°C was confirmed concomitantly. Library preparation of ≈550-bp insert size was performed with either Nextera or TruSeq nano HT kits (Illumina). Paired-end reads of 250 bp were generated for a pool of 10 barcoded Nextera libraries on a MiSeq instrument (Illumina, IBIS sequencing platform), and paired-end 100-bp reads were obtained

for a pool of 94 barcoded libraries on a HiSeq 2500 instrument (Illumina, Harvard sequencing platform).

Sequencing data were handled with Geneious 6.1.8 software (http://www.geneious.com/). Optimal number of mapping reads was achieved with a trimming of 0.01 for MiSeq reads and no trimming for the HiSeq reads. Two strategies were used to map reads. First, progenitor strains were mapped to their respective reference genome (S288c and RM11-1a) to build a progenitor genome for each genetic background. Default settings and 10 iterations were used. Unmapped paired reads were then *de novo* assembled, which lead to the identification of a 2 μm plasmid in the BY background and the recovery of the KanMX cassette. This plasmid and the KanMX cassette were included in the BY *las17-41* progenitor genome. The mitochondrial genome of S288c and the KanMX cassette were included in the RM *las17-41* progenitor genome. Evolved strains were then mapped to their progenitor genome using default setting and five iterations. Variant calling was performed with the following options: minimum of two reads and 0.75 frequency. An approximate *P*-value was computed for each variation. Variations at ambiguous positions in the progenitor genome were ignored. Polymorphism between progenitor strains was obtained by mapping RM *las17-41* reads on the BY *las17-41* progenitor genome with the same settings. Second, all reads were mapped to their reference genomes using the fastest settings and variants were called at 0.2 frequency, min 10× coverage, and a minimum *P*-value of 10e-10. The mutant strain reads were also mapped to their respective reference genomes using the default setting but with a max gap size of 60 bp and a max gap of 20% per read and five iterations. The variants were called at 0.75 frequency and a minimum *P*-value of $P = $ 10e-7, which was chosen based on capillary sequencing confirmations. Results were visualized directly in Geneious, and variant annotations were imported in JMP11 (SAS institute) for analysis.

**Identification of compensatory mutations**

To identify the most likely compensatory mutation for each mutant, we first looked for nuclear non-synonymous coding mutations. Sanger sequencing was used to validate non-synonymous coding mutations (CHUL sequencing platform) (Tables EV1 and EV4). Second, convergence outside of CDS was examined by binning mutation positions into 1-kb bins and focusing on bins where mutations were found in multiple mutant strains, but not in the progenitor strains. Third, to detect aneuploidy events, the average coverage of each contig/chromosome was estimated based on the number of mapping reads. Finally, because Sanger sequencing of candidate loci revealed some cases of transposon insertions and large deletions, the mappings of the remaining unexplained compensatory mutants were visually examined for clustered inconsistencies such as extreme distance between paired-ends and clustered mapping of unpaired reads. To confirm insertion events, reads were remapped to the progenitor genome keeping only reads mapping nearby. Unused paired reads were then subjected to *de novo* assembly, and contigs were annotated using BLASTn. In some cases, a confirmation PCR was also performed and when a PCR product could be obtained, capillary sequencing was performed to confirm the deletion or insertion (Table EV1). Compensatory changes in six mutants remain unidentified because of low or unequal sequencing coverage. In the few cases where more than one mutation was recovered

in a single strain, we conservatively assigned mutations as compensatory (Table EV1) or hitchhiker (Table EV4) based on the frequency of mutations in the same gene or functional target.

### Spot assays

Strains of interest were inoculated in SC glucose medium and grown overnight at 22°C. Each culture was adjusted to an $OD_{600\ nm}$/ml of one, diluted four times with a dilution factor of five, and 5 μl of each dilution was then spotted on the specified media and performed as described in Gagnon-Arsenault *et al* (2013).

### High-throughput growth assays of mutants

Growth of compensatory and reversion mutants along with progenitor and wild-type strains of each genetic background was assayed with a BM3-BC robot (S&P Robotics Inc., Canada). SC +G418 liquid pre-cultures were printed eight times in 384 arrays (four replicates per array, two replicate arrays), with border positions filled with a control strain. After 2 days of growth at 22°C, plates were replicated three times each and incubated for 3 days. Each plate was then used four times to print on synthetic (SC) and complex (YP) media containing glucose or galactose as a carbon source, and plates were incubated at 22, 30, and 37°C, and pictures were taken after 4 days. Images were analyzed using custom scripts written in ImageJ 1.45s (NIH). Growth was estimated by measuring colony sizes as described in Diss *et al* (2013). Values were normalized using the robust centered function in JMP and $\log_2$-transformed. Difference between each strain and their respective progenitor was obtained for each replicate and then averaged per strain.

### Ploidy measurements by flow cytometry

The same pre-cultures used for the high-throughput growth assays were used to inoculate SC media in either glucose or galactose at 22°C. Cells from a saturated culture were prepared as in Gerstein *et al* (2006) but stained with a final Sytox Green (Life Technologies Inc.) concentration of 0.6 μM for a minimum of 1 h. Fluorescence was measured using a guava easyCyte 8HT cytometer (Millipore) on up to 10,000 cells. Data were then analyzed using JMP11. DNA content is defined as the modal logarithmic green fluorescence value. The $k$-means function was used to define DNA content clusters. The cubic clustering criterion best fitted $k = 2$. Based on control samples, cluster assignments correspond to haploid and diploid genome sizes, and thus, strains are referred to as haploids and diploids.

### Complementation with compensatory alleles

Gibson cloning of selected alleles and their native promoter (1,000 upstream bp) was conducted using the backbone vector pRS316 which was first double digested with XbaI and BamHI. Primers and plasmids are listed in Table EV5. To minimize genetic manipulation of the thermosensitive strains and to facilitate testing alleles of essential genes, we introduced cloned alleles directly in the BY *las17-41* and RM *las17-41* thermosensitive strains. They were transformed with the constructed plasmids on SC –URA glucose +G418

and grown at 22°C. Transformations with pRS316 were performed in the thermosensitive strains and in wild-type strains as negative and positive controls, respectively. Using an automated procedure (BM3-BC robot), up to 24 transformed colonies of each transformation were arrayed in a 96-position format and grown for 2 days at 22°C. Plates were then replicated on SC –URA glucose +G418 and SC –URA galactose +G418 and incubated at 22 and 37°C. Pictures after 3 days of growth were used to assess complementation. Complementation for each colony was scored as positive if its size was superior to 95% of the appropriate control.

### Complementation by gene deletion

To delete genes in BY *las17-41* and RM *las17-41* thermosensitive strains, strains were first transformed with the plasmid pRSLAS17 containing the BY wild-type *LAS17* gene to restore Las17 function and to facilitate subsequent manipulations. pRSLAS17 was constructed according to Gibson cloning procedure as described, using *LAS17* amplified with its native promoter and terminator (1,000 upstream and downstream bp) (Table EV5). Genes of interest were then deleted in these strains and replaced with a natNT2 module amplified from pFA6-natNT2 plasmid (P30346, PCR toolbox, EUROSCARF—see Table EV3 for construction and confirmation primers) following a standard transformation procedure. Transformants were selected on SC –URA glucose +G418 +NAT. Finally, pRSLAS17 loss was induced in the confirmed deletion mutants by plating on SC –URA glucose medium supplemented with 1 g/l 5-FOA and 50 mg/l uracil.

### Cyclosporin A growth assays

To assess the compensatory effect of Cnb1 inhibition, spot assays were performed with parental and wild-type strains of both genetic backgrounds as well as BY *las17Δ* in the presence of the Cnb1 inhibitor cyclosporin A (Bioshop). Cyclosporin A was added to SC glucose and SC galactose media at final concentrations of 0, 1, 10, 25, 50 μg/ml, and 100 μg/ml. Plates were incubated at 22 and 37°C for 3 days.

### Determination of *las17* essentiality

Dissection of tetrads from BY *LAS17/las17Δ* and RM *LAS17/las17Δ* was performed on YP glucose medium following standard procedure. Germinated spores were plated on YP glucose +G418 to select for *las17Δ* genotype.

### Analysis

Prediction of the functional effect of all possible point mutations was computed with SNAP2 within the predict protein portal (Yachdav *et al*, 2014). Gene Ontology enrichments were obtained with the YeastMine widget (Balakrishnan *et al*, 2012). Protein–protein interactions were obtained from Biogrid 3.3.122 (Chatr-Aryamontri *et al*, 2015). The PPI enrichment was calculated by String 9.1 (Franceschini *et al*, 2013). Shortest paths were calculated using a custom R script using the igraph package where the *P*-value equals the probability score of the observed value in the distribution of 10,000 random samplings.

Association of compensatory genes with human disease pathways was performed with MORPHIN (Hwang *et al*, 2014) and the network visualized with Cytoscape 3.2.1 (Su *et al*, 2014). Statistical tests were performed with JMP11. Linear tests were used when the assumptions of the test were met (equal variance, normal distribution).

**Data availability**

Raw sequencing data are available at BioProject number PRJNA282519 at http://www.ncbi.nlm.nih.gov/bioproject/.

**Expanded View** for this article is available online:
http://msb.embopress.org

## Acknowledgements
This work was supported by Canadian Institute of Health Research (CIHR) grants 299432 and 324265 to CRL and a Human Frontier Science Program team grant (RGY0073/2010). CRL is a FRQS Junior II investigator and holds the Canada Research Chair in Evolutionary Cell and Systems Biology. We thank Charlie Boone for providing us with the thermosensitive allele collection. We thank Daniel Hartl and Nadia Aubin-Horth for comments on the manuscript.

## Author contributions
MF and CRL designed research; MF, VH, MCP, IGA, and AKD performed research; MF analyzed data; and MF, IGA, and CRL wrote the paper.

## Conflict of interest
The authors declare that they have no conflict of interest.

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
