## [Review Process File · Molecular Systems Biology]

Evolutionary rescue by compensatory mutations is constrained by genomic and environmental backgrounds

Marie Filteau, Véronique Hamel, Marie-Christine Pouliot, Isabelle Gagnon-Arsenault, Alexandre Dubé and Christian R Landry

*Corresponding author: Christian Landry, Université Laval -**

Review timeline:

Submission date:	14 July 2015
Editorial Decision:	21 August 2015
Revision received:	07 September 2015
Accepted:	14 September 2015

Editor: Maria Polychronidou

Transaction Report:

1st Editorial Decision

21 August 2015

Thank you again for submitting your work to Molecular Systems Biology. We have now heard back from two of the three referees who agreed to evaluate your manuscript. We are still expecting a report from reviewer #3, but since the recommendations of the other two referees are quite similar, I prefer to make a decision now rather than further delaying the process. If we receive comments from reviewer #3 we will forward them to you so that you can address any further issues raised. As you will see from the reports below, the referees think that the presented findings seem interesting. However, they raise a series of mostly minor concerns, which should be carefully addressed in a revision of the manuscript. The referees' recommendations are rather clear so there is no need to repeat all the points listed below.

Please resubmit your revised manuscript online, with a covering letter listing amendments and responses to each point raised by the referees. Please resubmit the paper ****within one month**** and ideally as soon as possible. If we do not receive the revised manuscript within this time period, the file might be closed and any subsequent resubmission would be treated as a new manuscript. Please use the Manuscript Number (above) in all correspondence. As a matter of course, please make sure that you have correctly followed the instructions for authors as given on the submission website.

REFeree REPORTS

Reviewer #1:

The authors analyze the rate and mechanism of genetic compensation of the high temperature growth defect caused by a mutation in the functional homolog of the WASP disease protein in yeast in two genetic backgrounds and in two carbon sources. They report that the rate of compensation,

the mutation mechanisms and the identity of the compensatory mutations change across conditions and genetic background.

The study is elegant, comprehensive and I find the results unexpected and thought provoking with respect to human genetic disease.

Suggestions:

It would be good to put the results in the context of what has already been shown about the plasticity of mutation effects and genetic interactions across conditions and genetic backgrounds.

There is previous work using reverse genetics showing that genetic interactions (primarily negative/aggravating interactions) can be sensitive to changes in environmental conditions (Harrison et al, PNAS 2007; St Onge et al, Nat Gen 2007; then later on a larger scale e.g. Bandyopadhyay et al, 2010; Guenole et al, 2013; Zhu et al, 2014), and the genetic backgrounds (actually between species (Tischler et al, 2008; Dixon et al, 2008; Roguev et al, 2008). A good example using natural genetic variation in yeast is the work on sporulation efficiency

Also the general result that single gene mutation effects are typically sensitive to the environment (e.g. the deletion collection in yeast) or genetic background (Boone, recent systematic analysis from Fraser in *C. elegans*) is also relevant for context. It would also be sensible to put the study slightly more in the context of recent systematic surveys of compensation e.g. from Pal.

The use of 'path' in the title is somewhat misleading as only single mutation steps underlie the compensation

It would be helpful to explain some jargon for the general reader e.g. 'higher order' interactions in the main text and perhaps the fluctuation analysis as a reminder in the methods section.

Reviewer #2:

I found the abstract to be unnecessarily vague, causing the first sentence of the paper to be rather jarring. I believe that the manuscript would be stronger if the abstract discussed the conclusions in a concrete way relative to *las17*. Moreover, I think that a paragraph in the beginning of the intro discussing the general issues and past work would be helpful before getting into the *las17* discussion. Finally, I think that the manuscript would benefit from more discussion of past results, both specifically regarding *las17* compensatory mutations and other research exploring context-dependency in evolution.

I think that the main text should have a brief explanation of the conditions in which the fluctuation assay was performed. In particular, since the authors have sequenced the resulting clones many readers will be interested in the basic setup (ie that the populations were grown at room temperature for three days before being plated on the non-permissive temperature of 37C). From the methods I didn't see some basic information that is necessary to interpret these results. What was the initial population size in each well? What was the number of divisions at room temperature over the three days of growth?

I found it quite interesting that the authors did not observe any mutations in *las17* other than in residue 41, which is where the mutation was introduced. I recommend that the authors include in the main text what fraction of the observed mutations at residue 41 were a reversion to the wildtype amino acid.

page 3: "Therefore the reversion rates cannot explain the context-dependent compensatory mutation rate." I think that this statement does not follow from the previous statement (but it is true). Only 5% of compensatory mutations were reversions, meaning that reversion rates cannot explain the context-dependent compensatory mutation rate independent of whether reversion rates are context-dependent.

Fig 1a: The y-axis is linear with spacing between adjacent ticks at 2×10^{-7} . However, the spacing

between the top two ticks is only 1×10^{-7} . This is presumably a simple error, but the authors should make sure that their manual processing of the figure / data did not introduce anything funny.

Regarding the right hand y-axis on diversity, I think that the authors need to provide some estimate of the error in their measurement so that the reader can decide whether the difference is statistically significant.

"...a missense suppressor mutation in the tRNA-Trp anticodon that restores the Las17 protein sequence by translation reading the arginine codon CGG as a tryptophan" I found this result to be very interesting. I would have thought that such a solution would have large fitness costs. If I am reading Fig 1b correctly, then this mutation indeed is costly at 22C for RM (but not BY). Oddly, it doesn't look like there is a fitness advantage at 37C... Is the more frequent observation of this mutation in the BY strain than the RM strain statistically significant? Frankly, I am somewhat surprised that it is observed so often in the RM strain given the apparently large fitness deficit.

Fig 1b: This is an important figure but I found it surprisingly difficult to extract the important points. Perhaps part of my confusion stemmed from the fact that I didn't know where the starting mutant would be on this plot so I couldn't judge which mutants were beneficial and which were not. A line plot is also likely not the best way to present the data, as it leads the reader to think that the growth rates for different mutations are somehow connected. Perhaps a stacked bar plot would make more sense.

I found the results on cyclosporine A and calcineurin inhibition to be quite interesting. I recommend that the authors simplify this figure for the main text and put the current version of the figure in the supplement. For example, they could just show the results in glucose and at 0, 10 ug/mL cyclosporine A. In addition, in the main text I recommend that they say that the positive effects of cyclosporine A are observed in glucose but not galactose (this weakens case) and only for the BY strain (this strengthens the case).

Figure 3B needs to be labeled better to explain that the "BY" and "RM" on the bottom horizontal axis refer to different alleles of the wild-type actin gene. The sentence in the main text referring to this figure is also very confusing, starting with the part that says "...most point mutations in actin." The second half of that sentence can only be inferred from Figure 3A, so the parenthetical should say "Figure 3a and 3b." Also, this sentence implies that the actin mutations were compensatory in glucose but not galactose in BY, but when I study the blue bars in 3A the pattern in BY is very similar for the two carbon sources.

Minor points:

I recommend that the authors use line numbers in the submitted draft to facilitate discussion of the paper.

In intro I would specify what the "classis WAS symptoms" are.

Mutation rate discussion: "We found a significant strain-by-carbon source effect..." I recommend that the authors specify the base result (ie that there appears to be a higher mutation rate for reversion of the phenotype in the RM / glucose combination)

1st Revision - authors' response

07 September 2015

Response to the referees

Reviewer comments - Author response

Reviewer #1:

1. It would be good to put the results in the context of what has already been shown about the plasticity of mutation effects and genetic interactions across conditions and genetic

backgrounds. There is previous work using reverse genetics showing that genetic interactions (primarily negative/aggravating interactions) can be sensitive to changes in environmental conditions (Harrison et al, PNAS 2007; St Onge et al, Nat Gen 2007; then later on a larger scale e.g. Bandyopadhyay et al, 2010; Guenole et al, 2013; Zhu et al, 2014), and the genetic backgrounds (actually between species (Tischler et al, 2008; Dixon et al, 2008; Roguev et al, 2008). A good example using natural genetic variation in yeast is the work on sporulation efficiency. Also the general result that effects are typically sensitive to the environment (e.g. the deletion collection in yeast) or genetic background (Boone, recent systematic analysis from Fraser in *C. elegans*) is also relevant for context. It would also be sensible to put the study slightly more in the context of recent systematic surveys of compensation e.g. from Pal.

We amended the introduction and discussion to include these suggestions as follow :

Line 9-29: “Genetic and environmental interactions constrain the course of adaptive evolution by limiting the number of available genetic paths to increased fitness (Chiotti et al, 2014; de Vos et al, 2013; Hartl, 2014; Kvitek & Sherlock, 2011; Weinreich et al, 2006). A given adaptive mutation may be advantageous in only a limited set of conditions, which makes its contribution to adaptation dependent on the environment and genotypes of the individuals in which the mutation travels on its way to fixation in the population. The effects of deleterious mutations are also variable across environments and genotypes, as illustrated by the variability of gene essentiality across conditions and genetic backgrounds (Dowell et al, 2010). In addition, these genetic (GxG) and genotype-by-environment (GxE) interactions can themselves be dependent upon other factors, revealing higher-order interactions (GxGxE, GxGxG and even, GxGxGxE). For instance, studies using reverse genetics showed that genetic interactions, primarily aggravating interactions, can be sensitive to changes in environmental conditions (GxGxE) (Bandyopadhyay et al, 2010; Guenole et al, 2013; Harrison et al, 2007; St Onge et al, 2007; Zhu et al, 2014), and also diverge between related and distant species (GxGxSp) (Dixon et al, 2008; Roguev et al, 2008; Tischler et al, 2008). These complex interactions suggest that overcoming the effect of a deleterious mutation by compensatory evolution is most likely subjected to the same constraints as adaptive evolution, as highlighted by a recent report on plastic compensatory mutation effects across environments (GxGxE) (Szamecz et al, 2014). Understanding these complex interplays of genetic and environmental effects is one of the great challenges in systems biology (Fischbach & Krogan, 2010; Weinreich et al, 2013) because it requires a mechanistic understanding of genotype-phenotype maps (Landry & Rifkin, 2012).”

Line 194-197: “However, although the BY and RM *las17-41* strains had a similar phenotype, their respective genetic network underpinning the actin cytoskeleton regulation appears to have diverged substantially since they last shared a common ancestor, providing a prime example of developmental system drift (True & Haag, 2001; Verster et al, 2014).”

2. The use of 'path' in the title is somewhat misleading as only single mutation steps underlie the compensation

We changed the title to:

“Evolutionary rescue by compensatory mutations is constrained by genomic and environmental backgrounds”

3. It would be helpful to explain some jargon for the general reader e.g. 'higher order' interactions in the main text and perhaps the fluctuation analysis as a reminder in the methods section.

We introduced the following explanations:

Line 16-18: “In addition, these genetic (GxG) and genotype-by-environment (GxE) interactions can themselves be dependent upon other factors, revealing higher-order interactions (GxGxE, GxGxG and even, GxGxGxE).”

Line 72-78: “The fluctuation assay is a short-term evolution experiment in which small parallel populations of the thermosensitive strains are allowed to accumulate conditionally neutral mutations in synthetic media under permissive conditions (22 °C) for around 20 generations. The genetically variable populations are then placed under selective conditions (37 °C). Only the rare cells that pre-

acquired a rescue mutation are able to form a colony and the frequency of these events is recorded at the population level to calculate the compensatory mutation rate.”

Reviewer #2:

4. I found the abstract to be unnecessarily vague, causing the first sentence of the paper to be rather jarring. I believe that the manuscript would be stronger if the abstract discussed the conclusions in a concrete way relative to *las17*.

We provided a proper abstract in accordance to the journal guidelines.

Abstract: “Because deleterious mutations may be rescued by secondary mutations during evolution, compensatory evolution could identify genetic solutions leading to therapeutic targets. Here, we tested this hypothesis and examined whether these solutions would be universal or would need to be adapted to one’s genetic and environmental makeups. We performed experimental evolutionary rescue in a yeast disease model for the Wiskott-Aldrich Syndrome in two genetic backgrounds and carbon sources. We found that multiple aspects of the evolutionary rescue outcome depend on the genotype, the environment or a combination thereof. Specifically, the compensatory mutation rate and type, the molecular rescue mechanism, the genetic target and the associated fitness cost varied across contexts. The course of compensatory evolution is therefore highly contingent on the initial conditions in which the deleterious mutation occurs. In addition, these results reveal biologically favored therapeutic targets for the Wiskott-Aldrich Syndrome, including the target of an unrelated clinically approved drug. Our results experimentally illustrate the importance of epistasis and environmental evolutionary constraints that shape the adaptive landscape and evolutionary rate of molecular networks.”

5. Moreover, I think that a paragraph in the beginning of the intro discussing the general issues and past work would be helpful before getting into the *las17* discussion.

See response to comment #1.

6. Finally, I think that the manuscript would benefit from more discussion of past results, both specifically regarding *las17* compensatory mutations and other research exploring context-dependency in evolution.

We included the following text to discuss this topic:

Line 189-107: “When applied to *LAS17*, this approach brings further insights into the essential role of Las17/WAS WH1 domain for which little is known beyond its binding affinity to Vrp1/WIP (Cotta-de-Almeida et al, 2015; Zettl & Way, 2002), and highlights connections to other immune related homologs. The results reveal potential therapeutic targets for pharmaceutical modulation of WASP-mediated signalling, which could evolve as a rational strategy. However, although the BY and RM *las17-41* strains had a similar phenotype, their respective genetic network underpinning the actin cytoskeleton regulation appears to have diverged substantially since they last shared a common ancestor, providing a prime example of developmental system drift (True & Haag, 2001; Verster et al, 2014).”

7. I think that the main text should have a brief explanation of the conditions in which the fluctuation assay was performed. In particular, since the authors have sequenced the resulting clones many readers will be interested in the basic setup (ie that the populations were grown at room temperature for three days before being plated on the non-permissive temperature of 37C). From the methods I didn't see some basic information that is necessary to interpret these results. What was the initial population size in each well? What was the number of divisions at room temperature over the three days of growth?

See response to comment #3. We also added more details in the method section:

Line 233-239: “For each experiment, a thermosensitive colony was diluted in 40 mL of SC medium with 2% glucose or galactose. The number of cell per uL was estimated three times by flow cytometry with a guava easycyte 8HT cytometer (Milipore) and 30 μ L aliquots containing around 5,000 cells each were placed in 46 wells of a 96-well PCR plate, avoiding corner wells. After three days of incubation at room temperature (~20 generations), the whole content of 40 wells was manually spotted on an omnitray and placed at 37 °C for seven days.”

8. I found it quite interesting that the authors did not observe any mutations in *las17* other than in residue 41, which is where the mutation was introduced. I recommend that the authors include in the main text what fraction of the observed mutations at residue 41 were a reversion to the wildtype amino acid.

We included the proportions as follow:

Line 89-90: “All intramolecular mutations affected the originally mutated residue and we encountered both true reversions (R41W, 11%) and pseudo-reversions (R41L, 44%, R41Q, 44%).”

9. page 3: "Therefore the reversion rates cannot explain the context-dependent compensatory mutation rate." I think that this statement does not follow from the previous statement (but it is true). Only 5% of compensatory mutations were reversions, meaning that reversion rates cannot explain the context-dependent compensatory mutation rate independent of whether reversion rates are context-dependent.

We relocated the sentence as follow:

Line 97-89 : “Only 5% of compensatory mutations occurred within *las17-41* itself. The reversion rates can therefore not explain the context-dependent compensatory mutation rate.”

10. Fig 1a: The y-axis is linear with spacing between adjacent ticks at $2 \cdot 10^{-7}$. However, the spacing between the top two ticks is only $1 \cdot 10^{-7}$. This is presumably a simple error, but the authors should make sure that their manual processing of the figure / data did not introduce anything funny.

Thank you for catching this mistake. This is indeed an error introduced during manual processing of the figure. The mistake has been corrected.

11. Regarding the right hand y-axis on diversity, I think that the authors need to provide some estimate of the error in their measurement so that the reader can decide whether the difference is statistically significant.

We used this measure to illustrate the lack of correlation with the compensatory mutation rate, to show that the diversity of solution (accounting for both the number of solutions and their frequency), cannot explain the difference in mutation rate. Although the identified mutations come from repeated fluctuation assays, the sample sizes are too uneven to compute a diversity index individually for each fluctuation assay. Therefore, to simplify the matter, we replaced the Shannon index by the number of unique mutation found in each of the four experiments. Accordingly, we modified the text as follow:

Line 100-103: “The number of locus-unique mutations did not correlate with the compensatory mutation rate but differed substantially among contexts, with the largest difference observed between genetic backgrounds in galactose (**Fig 1A**). Thus, the elevated mutation rate in the RM-glucose condition is apparently not owed to a broader mutational target size of rescue mutations.”

Line 413-414: “Overlaid dots show the number of unique rescue mutation (right axis) found in each context. Mutations at a shared locus and nucleotide position were considered equivalent.”

12. "...a missense suppressor mutation in the tRNA-Trp anticodon that restores the Las17 protein sequence by translation reading the arginine codon CGG as a tryptophan" I found this result to be very interesting. I would have thought that such a solution would have large fitness costs. If I

am reading Fig 1b correctly, then this mutation indeed is costly at 22C for RM (but not BY). Oddly, it doesn't look like there is a fitness advantage at 37C... Is the more frequent observation of this mutation in the BY strain than the RM strain statistically significant? Frankly, I am somewhat surprised that it is observed so often in the RM strain given the apparently large fitness deficit.

The difference in frequency of the tRNA-Trp mutants across experimental conditions is indeed statistically significant. Because the rescue experiment is not an adaptation experiment, the selection against high fitness cost mutation is not effective, which allows us to sample those mutations as well. In the fluctuation assay, the whole population of cells in each parallel culture is placed in the selective condition. Therefore, as long as the mutation is not lethal, it can be recovered in this experiment if it allows the cell to grow in the selective condition. The fluctuation assay is designed in such a way that compensatory mutations are rare enough that they occur only in a fraction of the populations and most of the time, only one mutant per population is observed and can thus be sampled without bias. Furthermore, the tRNA-Trp gene is present in multiple copies in the yeast genome, increasing the probability of observing this mutation which as been clarified in the text:

Line 105-112: “The most frequent mutation (36 total occurrences) was a suppressor mutation in the tRNA-Trp anti-codon that restores the Las17 protein sequence by translating the arginine codon CGG as a tryptophan (**Table EV1**). An increased number of targets can explain the high frequency of this mutation, as there are six nuclear copies of the tRNA-Trp gene in the yeast genome, in each of which we observed individual occurrences of the same suppressor mutation. Interestingly, the frequency of this solution varied between contexts (nominal logistic model, $df = 3$, whole model P value = 0.005, interaction P value = 0.005) (**Fig 1C**), again indicating complex interactions in the favored evolutionary paths.”

13. Fig 1b: This is an important figure but I found it surprisingly difficult to extract the important points. Perhaps part of my confusion stemmed from the fact that I didn't know where the starting mutant would be on this plot so I couldn't judge which mutants were beneficial and which were not. A line plot is also likely not the best way to present the data, as it leads the reader to think that the growth rates for different mutations are somehow connected. Perhaps a stacked bar plot would make more sense.

We modified figure 1b to better reflect the rescued fitness and cost compared to the progenitor strain.

Fig 1B legend, line 416-422: “The average growth of evolved strains sharing compensatory mutation types at 22 °C (top panel) and 37 °C (bottom panel) in four culture media is shown relative to their respective progenitor. The *las17* strains carry R41W, R41L or R41Q functional reversions. WT = wild type strain. Growth phenotypes were measured on synthetic complete (SC) and rich media (YP), n = number of strains next to labels, each inferred from eight replicates. One *cap1* mutant was excluded because of mitochondrial loss and one mutant containing two rescue mutations (*act1* and *cap2*).”

14. I found the results on cyclosporine A and calcineurin inhibition to be quite interesting. I recommend that the authors simplify this figure for the main text and put the current version of the figure in the supplement. For example, they could just show the results in glucose and at 0, 10 ug/mL cyclosporine A. In addition, in the main text I recommend that they say that the positive effects of cyclosporine A are observed in glucose but not galactose (this weakens case) and only for the BY strain (this strengthens the case).

We proceeded as suggested for the figure. As for the effects of cyclosporine and *cnb1* deletion being specific to glucose, we would like to argue that we only found one mutant in galactose with a *cnb1* mutation, against 6 in glucose, therefore the cyclosporine effect matches the rescue mutants frequency quantitatively rather than qualitatively. Because the rescue mutations were transposon insertions, it is possible that the mechanism of compensation differs from complete inactivation of the gene or alternatively, a second undetected rescue mutation could be present in this strain.

we modified the text as follow:

Line 178-181: “We found that cyclosporin A allows growth of BY *las17-41* at 37 °C on glucose (**Fig 6 and EV5**) which mirrors the *las17-41 cnb1D* phenotypes (**Fig EV4**) and the quantitative results obtained for *cnb1* rescue mutants (**Fig 1C**).”

15. Figure 3B needs to be labeled better to explain that the "BY" and "RM" on the bottom horizontal axis refer to different alleles of the wild-type actin gene. The sentence in the main text referring to this figure is also very confusing, starting with the part that says "...most point mutations in actin." The second half of that sentence can only be inferred from Figure 3A, so the parenthetical should say "Figure 3a and 3b." Also, this sentence implies that the actin mutations were compensatory in glucose but not galactose in BY, but when I study the blue bars in 3A the pattern in BY is very similar for the two carbon sources.

Figure 3B has been amended to include the label “WT *ACT1* allele”. Also, there was indeed a writing mistake, we modified the sentence accordingly :

Line 153-157: “The actin gene is a prime example supporting this conclusion, as increasing gene-dosage of actin with a wild type allele was able to compensate *las17-41* thermosensitivity specifically in the RM background on glucose, whereas some point mutations in actin were compensatory in all contexts (**Fig 4A and 4B**).”

16. In intro I would specify what the "classic WAS symptoms" are.

We added the following information:

Line 47-49: “WAS is a rare X-linked primary immunodeficiency and blood platelet disorder classically characterized by the triad of recurrent infections, abnormal bleeding caused by a reduced number of platelets, and skin eczema (Albert et al, 2011).”

17. Mutation rate discussion: "We found a significant strain-by-carbon source effect..." I recommend that the authors specify the base result (ie that there appears to be a higher mutation rate for reversion of the phenotype in the RM / glucose combination)

We added the following sentence:

Line 81-82: “Indeed, there is a higher mutation rate for the ability to rescue growth at 37 °C in the RM genetic background on glucose (Tukey HSD, P value = <0.05).”